# Deep Multi-Objective Learning from Low-Dose CT for Automatic Lung-RADS Report Generation

**DOI:** 10.3390/jpm12030417

**Published:** 2022-03-08

**Authors:** Yung-Chun Chang, Yan-Chun Hsing, Yu-Wen Chiu, Cho-Chiang Shih, Jun-Hong Lin, Shih-Hsin Hsiao, Koji Sakai, Kai-Hsiung Ko, Cheng-Yu Chen

**Affiliations:** 1Graduate Institute of Data Science, Taipei Medical University, Taipei 110, Taiwan; changyc@tmu.edu.tw (Y.-C.C.); m946106006@tmu.edu.tw (Y.-C.H.); linda037@tmu.edu.tw (Y.-W.C.); 2Clinical Big Data Research Center, Taipei Medical University Hospital, Taipei 110, Taiwan; 3Center for Big Data and Artificial Intelligence in Medical Imaging, Taipei Medical University, Taipei 110, Taiwan; cc.shih@tmu.edu.tw (C.-C.S.); junhonglin@tmu.edu.tw (J.-H.L.); 4Division of Pulmonary Medicine, Department of Internal Medicine, School of Medicine, College of Medicine, Taipei Medical University, Taipei 11031, Taiwan; hsiaomd@gmail.com; 5Division of Pulmonary Medicine, Department of Internal Medicine, Taipei Medical University Hospital, Taipei 11031, Taiwan; 6Department of Radiology, Kyoto Prefectural University of Medicine, Kyoto 602-8566, Japan; sakai3@koto.kpu-m.ac.jp; 7Department of Radiology, Tri-Service General Hospital and National Defense Medical Center, Taipei 114, Taiwan; m860818@gmail.com; 8Department of Medical Imaging, Taipei Medical University Hospital, Taipei 110, Taiwan; 9Translational Imaging Research Center, Taipei Medical University Hospital, Taipei 110, Taiwan; 10Department of Radiology, School of Medicine, College of Medicine, Taipei Medical University, Taipei 110, Taiwan; 11Research Center for Artificial Intelligence in Medicine, Taipei Medical University, Taipei 110, Taiwan

**Keywords:** natural language processing, automatic radiology report generation, deep neural network, medical informatics

## Abstract

Radiology report generation through chest radiography interpretation is a time-consuming task that involves the interpretation of images by expert radiologists. It is common for fatigue-induced diagnostic error to occur, and especially difficult in areas of the world where radiologists are not available or lack diagnostic expertise. In this research, we proposed a multi-objective deep learning model called CT2Rep (Computed Tomography to Report) for generating lung radiology reports by extracting semantic features from lung CT scans. A total of 458 CT scans were used in this research, from which 107 radiomics features and 6 slices of segmentation related nodule features were extracted for the input of our model. The CT2Rep can simultaneously predict position, margin, and texture, which are three important indicators of lung cancer, and achieves remarkable performance with an F_1_-score of 87.29%. We conducted a satisfaction survey for estimating the practicality of CT2Rep, and the results show that 95% of the reports received satisfactory ratings. The results demonstrate the great potential in this model for the production of robust and reliable quantitative lung diagnosis reports. Medical personnel can obtain important indicators simply by providing the lung CT scan to the system, which can bring about the widespread application of the proposed framework.

## 1. Introduction

The global burden of cancer morbidity and mortality is increasing rapidly. Lung cancer is the most common cancer, the leading cause of cancer deaths in men, and the second leading cause of cancer deaths in women [1]. Among lung cancer patients diagnosed between 2010 and 2014, the 5 year survival rate of lung cancer patients in most countries is only 10% to 20% after diagnosis [2]. Low-dose computed tomography (LDCT) is used for high-risk groups and can help diagnose cancer early when it is more likely to be successfully treated. The efficacy of low-dose CT screening once a year in reducing lung cancer mortality has been confirmed in many independent international randomized controlled clinical trials [3,4]. The category of the tumor is further diagnosed by referring to the Lung-RADS Version 1.1 (https://www.acr.org/-/media/ACR/Files/RADS/Lung-RADS/LungRADSAssessmentCategoriesv1-1.pdf, accessed on 30 September 2020). Application of Lung-RADS report after clinical CT scans can increase the positive predictive value and assist decision making [5]. Extracting information from LDCT scans and converting it into medical reports can be time-consuming. However, if the time can be shortened, the quality and effectiveness of medical care can be improved. In light of this, we aim to interpret from LDCT scans to generate a text report with Lung-RADS by utilizing machine learning methods.

Extracting information from chest computed tomographic (CT) scans and converting it into medical reports can be time-consuming for radiologists. This major burden has in recent years driven researchers to develop automatic report generation methods, especially automatic diagnosis systems for tumors, which are booming in recent years. For instance, the authors of [6] improved and reduced the measuring time for the classic Raman micro-spectroscopy detection method, which could effectively detect skin cancer. Reference [7] developed an automatic prostate cancer detection method, based on a learning-based multi-source integration framework, that was found to perform better on cancer localization than four conventional approaches. Furthermore, the authors of [8] introduced a new approach to overcome the problem of unbalanced data distribution on brain tumor classification, which always poses considerable challenges for clinical experiments and research. These recent works have all contributed to the rapid development in automatic cancer diagnosis, and the current paper further extends this research direction.

Moreover, with continuous breakthroughs in deep learning models for image recognition continue [9], automatic diagnoses of tumors have received more and more attention. Consider the following examples: References [10,11] presented several deep reinforcement learning models with great potential for detecting lung cancer. Reference [12] proposed a new method to segment brain tumor images that is more accurate and effective than other methods, and the authors of [13] introduced a classifier that can distinguish four classes of brain tumors by using deep learning models with discrete wavelet transform (DWT) and principal components analysis (PCA) techniques. In addition, several recent studies adopted deep neural networks for automatic diagnostic image analysis. AlexNet was utilized to detect lung abnormalities in [14], and a 2D CNN was employed to detect lung nodules in [15]. DeepLung [16] was developed for automatic pulmonary nodule detection from 3D CT scans using deep 3D convolution neural networks. Although deep learning is widely applied to automatic tumor diagnosis, it is unfortunate that an automated lung radiology report generation system has yet to be developed, even though recent studies have proposed methods that can effectively extract cancer information from lung CT. Therefore, it will greatly assist radiologists if these methods can be used to generate radiology reports.

In terms of clinical aspects, information extraction has gradually become a clinical aid, and information extraction mainly comes from natural language processing (NLP) technology. For instance, Hunter et al. [17] used NLP technique to automatically generate natural language nursing shift summaries from electronic medical records, which is very effective in assisting clinical decision making and report generation. Moreover, generating coherent radiology paragraphs has been the subject of recent academic attention, but there are few research articles on this subject [18]. Xue et al. [19] proposed a multimodal recurrent model to generate radiology report paragraphs using chest X-rays as part of a computer-aided reporting system. The results demonstrate that their system can help clinicians make timely and effective decisions to solve problems of staff shortage and excessive workload. Furthermore, Wang et al. [20] proposed the text–image embedding network (Tie-Net), which is a joint learning model that integrates convolutional neural network and recurrent neural network with attention mechanism to classify chest X-ray images and generate preliminary reports.

In light of this, we aimed to develop a comprehensive system for automatic lung radiology report generation. In this paper, we propose a multi-objective neural network model, named CT2Rep, that can predict three important semantic features (i.e., location, texture, and margin) of tumors from CT scans. The category of the tumor is further diagnosed by referring to the Lung-RADS Version 1.1. We designed a set of linguistic templates to encapsulate the syntax and context within lung radiology reports. Finally, a structured textual radiology report was generated by summarizing the nodule level through the Lung-RADS model and filling the slots in templates. The experimental results based on real-world datasets demonstrated that the proposed CT2Rep is effective in semantic feature prediction, outperforming numerous well-known machine learning and deep neural network methods. Furthermore, the generated radiology reports are deemed to be clinically serviceable according to satisfaction assessment by professional radiologists.

## 2. Materials and Methods

### 2.1. Data Acquisition

The data collected from actual patients greatly contributed to the importance and credibility of this research. First of all, regarding the legality of data acquisition, this study was approved by the Joint Institutional Review Board of the Taipei Medical University Hospital (TMUH) (IRB No. N202202003). The records were collected by the TMU Office of Data Science (TMUODS) from patients with final pathological confirmation or clinical diagnosis. TMUODS has been integrating electronic medical records of three affiliated hospitals of Taipei Medical University to form the Taipei Medical University Clinical Research Database (TMUCRD). With the establishment of the TMUODS and after obtaining the consent of the patients, Taipei Medical University de-identified the clinical data from these affiliated hospitals and constructed this clinical database. At the same time, it also formulated database management methods and application procedures to standardize the management of the database, access, usage rules, and security protocol, as well as to enable health data to be used for biomedical research. To avoid data bias, these patients were reviewed retrospectively from 2016 to 2019, collected from TMUH, Taipei Medical University Shuang Ho Hospital, and Taipei Municipal Wanfang Hospital. During the data collection period, approximately 400 people passed away, and 100 people opted out. In order to filter out the target samples for this research, CT reports of target patients were gathered by initially searching for keywords “CT” and “nodule”. After this, CT report cases with the keywords “nodule”, “opacity”, “GGO (ground-glass opacity)”, “adenocarcinoma”, “granuloma”, “metastasis”, and “cancer” in the “Type” section, and “pleural”, “hilum”, “pulmonary”, “lung”, “RUL (right upper lobe)”, “RLL (right lower lobe)”, “RML (right middle lobe)”, “LLL (left lower lobe)”, and “LUL (left upper lobe)” in the “Position” section were collected. It is worth noting that the CT reports with “shadow,” “emphysema,” “pneumonia”, “pneumonitis”, “cysts”, “fibrotic foci”, “inflammation”, and “consolidative patchy” were excluded due to the lack of relevance to this research.

To acquire the image of target patients, the inclusion criteria of our study were as follows: (1) the patient used a 5 mm thick conventional CT scan, (2) the diagnosis of no distant metastasis was confirmed by surgery and pathology, (3) surgery or biopsy was only selected before the last CT scan, (4) the diameter of each nodule was less than 30 mm, (5) only single-nodule lung CT scans are included. According to the above criteria, this study included a total of 456 cases (220 women and 236 men, with an average age of 65 ± 13), which contained 458 cases of lung nodules overall. In the process of obtaining the data, all patients took chest CT scans in the supine position on the scanning bed under free-breathing conditions. The CT scanners were produced by three manufacturers (GE Medical Systems, Philips Medical Systems, and Siemens) and were used to collect CT scans of 110–120 kV and 10–20 mA. The image slice matrix was 512 × 512, the slice thickness was 5 mm, and the pixel pitch was 0.618 × 0.618 mm^2^.

The semantic features of nodule morphology that are essential to the report generation process were defined through an in-depth discussion and comprehensive evaluation among three radiologists with 10 to 20 years of radiology experience. In the end, three semantic features were selected as the target for the machine learning model in this re-search. The first semantic feature is “Location”, which contains six types of nodule locations, namely, right upper lobe (RUL), right middle lobe (RML), right lower lobe (RLL), left upper lobe (LUL), left lower lobe (LLL), and Lingular Lobe, which is slightly covered by the LUL. These semantic labels represent the position of the nodule in the entire lung. The second is “Texture”, which contains three semantic labels about the attenuation of the nodule, i.e., Solid, Subsolid, and Pure GGO, listed from higher to lower density. The last semantic feature is “Margin”, with four semantic labels, i.e., Sharp Circumscribed, Lobulated, Indistinct, and Spiculated. These semantic labels are related to the border of the nodule and whether the tumor is malignant. Notably, the last two categories of semantic features represent the internal texture of the nodule and the boundary of the nodule. For this reason, they are also called intratumoral semantic features.

These CT findings were analyzed on the basis of the lung window of the CT scan (with a Hounsfield unit (HU) value range of −1400 to 400 HU). In addition, the semantic features extracted from the nodules were recorded on the Radiology Information System (RIS). The semantic features of the nodules were designed as checkable items, including texture, margin, and position on the RIS. Each of these three characteristics has its own subtypes or labels, as shown in Table 1. Finally, all target nodules were examined and reviewed twice by different radiologists.

### 2.2. Radiomics Feature Extraction

Given a lung CT, we first segmented the nodule to increase the precision of the extracted image features. We utilized the semi-automatic segmentation method IntelliSpace Discovery (https://www.philips.com.tw/healthcare/product/HC881015/intellispace-discovery/gongying, accessed on 18 September 2020) to segment all target nodules and then stored them in a DICOM format. After this, the segmented contour was checked again by another doctor, and if necessary, the contour of the nodule was further modified by freehand drawing. Next, since [21,22] mentioned radiomics features show high accuracy for classification features and high assistance in building clinical decision system, we employed Pyradiomics [23] to obtain our aspired radiomics features from the nodule image in control of the segmentation outcomes. The process of Pyradiomics extracting radiomics features from medical imaging is as follows. First, several filters (e.g., Laplacian of Gaussian and Wavelet) can be adapted to remove the noise of the images. Next, it proceeds to the core part, which is the calculation of radiomics features. As shown in Table 2, a total of 107 radiomics features were extracted and classified into two main groups: first-order and second-order statistics [24]. The first-order features are related to the characteristics of intensity distribution in the VOI and the shape-based 2D and 3D morphological features of the VOI as well. On the other hand, the second-order features can be seen as a textural analysis, providing a measure of intra-lesion heterogeneity and further assessing the relationships between the pixel values within the VOI. These include gray level co-occurrence matrix (GLCM), gray level run length matrix (GLRLM), gray level size zone matrix (GLSZM), neighboring gray-tone difference matrix (NGTDM), and gray level dependence matrix (GLDM). In addition, we found that the ratio of labeled slices to total slices and some other related slicing information indicated the location of the nodules to a certain extent. Therefore, a total of six features were extracted from the slice information of segmentation of nodules (SISN) and were used in our proposed method.

To summarize, our dataset consists of a total of 458 labeled imaging data, from which 113 features were extracted (107 radiomics features and 6 features generated from the SISN). These data serve as the initial training data for the proposed CT2Rep method.

### 2.3. CT2Rep: An Effective Multi-Objective Learning Method for Radiology Report Generation

In this section, we present the details of the CT2Rep, an automatic generation method for radiology reporting from lung CT scan through semantic feature prediction. After the Radiomics Feature Extraction process in the previous section, we are able to obtain a 113-dimensional image feature. Thereafter, the proposed multi-objective neural network is used to predict the semantic features from the lung CT, which are the basis for inferring the Lung-RADS. Finally, the radiology report of the lung CT is generated through the slot filling procedure of the linguistic templates. Figure 1 illustrates the architecture of the proposed CT2Rep model. Given a 113-dimension radiomics feature vector, a neural network is trained to predict the semantic features that are important factors of writing a radiology report, i.e., the Location, Texture, and Margin. Initially, only 107 of the 113-dimensional radiomics features are required for predicting the Margin and Texture; however, the inference of Location requires not only the 107-dimensional radiomics features but also 6 of the extra SISN features extracted from the information of nodule segmentation.

Since the input to CT2Rep is a set of features in vector form extracted from VOI of lung CT scans rather than raw pixels from the image, the convolutional neural network-based method may be less suitable as a classifier [25]. In addition, this type of data contains no temporal attribute or sequential relationship between data points, which means that recurrent neural network-based approaches are not suitable for building our classifier [26]. As mentioned in [27,28], multilayer perceptron (MLP) is appropriate for input features with both linguistic and numerical types. Moreover, paired with other neural network-related technologies such as backward propagation [29], activation functions, and deep hidden layers, the MLP becomes a strong classifier. Reference [30] also proposed that the advantage of MLP is that it can make more accurate predictions compared to traditional classification algorithms without giving weights or making decisions about the relative importance of various input features. Therefore, we adopt MLP as the fundamental neural structure of CT2Rep. More specifically, we model Margin and Texture using the following architecture.

The merged 113-dimension radiomics features as dense vectors for input to a neural network. Afterward, these three categories of features are learned by separate stacks of multiple dense layers, which constitute the MLP. Two dense layers are adopted for extracting latent features for predicting semantic labels. In the fully connected neural network, a neuron is a computational unit that has scalar inputs and outputs. As shown in Equation (1), the neuron *n_i_* sums the multiplication of input *x_j_* by its weight *w_j_* with bias *b_j_*, and a non-linear activation function σ(z) is utilized to the result for output $\hat{y_{i}}$ The output of a neuron may feed into the inputs of one or more neurons, and the neurons are connected to each other forming a fully connected layer. The activation functions ReLU [31] and tanh [32] are utilized for Margin and Texture, respectively; for the prediction of Location, we use the tanh function.
(1)$$\hat{y_{i}}=\sigma \left(\underset{j}{\sum }w_{j}x_{j}+b_{j}\right)$$

The last procedure in our model is to construct a classifier to predict the semantic la-bels. The softmax function is adopted for normalization, and the approximate probability of each semantic category is calculated from the output of the fully connected layer. This completes the primary mission of CT2Rep: to predict multiple semantic features from CT scans. The calculation formula is as follows.
(2)$$\hat{y}=softmax\left(MF^{T}+b\right)$$
where *M* is the parameter matrix of the connection layer, *F* is the characterization of the distributed characteristics, *b* is the bias, and softmax is a normalization function. Moreover, due to the limited number of samples, we also exploit dropout layers [33,34] in between the dense layers to prevent the phenomenon of over-fitting. The dropout will randomly stop some neurons from propagating their output to the next layer, which is to say they are shielded. This can increase the generalization ability of the network, which means that the model is more likely to extract universal features rather than focus on very specific ones that may only exist in this training data. For the prediction of Location, our dropout rates were set as 0.35 and 0.25 from the input side to the output, while the drop-out rates of the prediction for Texture were 0.25 and 0.15. In addition, RMSprop optimizer [35] is utilized to optimize the loss function of the network, and the parameters can thus be fine-tuned effectively through the backpropagation mechanism. Our loss function is cross-entropy due to it being able to reduce the risk of a gradient disappearance during the process of stochastic gradient descent, and thus it is often better than the classification error rate or the mean square error [36]. The loss rate of the model can be calculated using the following equation:(3)$$Loss=-\overset{N}{\underset{i=1}{\sum }}y_{i}\times \log\hat{y_{i}}+\left(1-y_{i}\right)\times \log\left(1-\hat{y_{i}}\right)$$
where *N* is the number of training samples, *y* is the label of the sample, and $\hat{y}$ is the output of the model. The multi-objective MLP model is implemented using Keras (https://keras.io/, accessed on 24 September 2020). Regarding the implementation details, the batch size was set at 32, and the training loop lasted for 50 epochs.

Subsequently, we included Lung-RADS for the report generation module, since it has in recent years become an important indicator of lung cancer [5,37]; as shown in Figure 2, we used the standards of Lung-RADS Assessment Categories Version 1.1, from the American College of Radiology (https://www.acr.org/Clinical-Resources/Reporting-and-Data-Systems/Lung-Rads, 30 October 2020), to formulate the decision process of the Lung-RADS. Once the semantic labels are predicted, the Lung-RADS of the lung CT can be obtained through this process with the predicted semantic labels and extracted radiomics features altogether. After this, the above information is automatically converted into a structured radiology report through the text generation mechanism, which is based on linguistic templates because the content of the report must be highly controllable. In addition, even though a large training dataset is re-quired when developing a text generation model, for medical data and especially labeled CT scans, flawless data are in practice very rare. Thus, a template-based approach is more suitable in this scenario. To develop well-formed linguistic templates for report generation, we started from the templates provided by the RSNA website, and then consulted with professional radiologists in order to refine them. In this way, a proficient radiology report can be generated through slots filling of the predicted semantic features and Lung-RADS information. By connecting a series of modules, radiology reports can be automatically generated from a lung CT scan.

Lung-RADS is more subjective, and therefore we recommended the use of rule-based methods to complete its mechanism and combining it with human–computer interaction, such as modifying the texture category, to tune the classes. In real world data, the distribution of Lung-RADS is also very unbalanced like the data we collected, making it difficult to train the model; on the other hand, the version of Lung-RADS will also be updated from time to time. The classification method using Rule is relatively free and easy to maintain for the aspect of a product. When the Lung-RADS is upgraded, we can immediately follow the revision.

As previously stated, one of the purposes of this framework is to help reduce the workload of radiologists. Therefore, we also built a prototype system in which the radiologists can review and verify these preliminary radiology reports instead of writing a new one from scratch every time. In this way, the burden on radiologists will be greatly alleviated, and they can pay more attention to actually examining the images. In order to estimate the practicality of the system for clinical professionals, we conducted a satisfaction analysis experiment to assess the acceptance of the CT2Rep system by radiologists. Two professional radiologists specializing in radiology participated in the satisfaction survey. They scored the text reports generated by CT2Rep, which were based on 100 cases randomly sampled from our dataset using 5-point scores on a Likert Scale for (1) the match between CT scans and the generated reports, and (2) the fluency and logic of the text.

## 3. Results

### 3.1. Semantic Feature Prediction

The performance evaluation metrics used in this research are precision, recall, and F_1_-score [38], which are defined in Equations (4)–(6). The *TP* indicates the number of true positives, which are the number of positive instances that are correctly classified. The *FP* represents the number of false positives, which are negative instances that are erroneously classified as positives. Similarly, *TN* and *FN* stand for the number of true negatives and false negatives, respectively. Considering there is a trade-off between precision and recall due to both metrics, we evaluated system performance from different perspectives. F_1_-score is the harmonic mean of precision and recall, which can thus be considered as an attempt to find the best possible compromise between precision and recall [38]. For this reason, F_1_-score is used to judge the superiority of comparisons. We used macro-average to compute the average performance. To derive credible evaluation results, we employed the leave-one-out cross-validation approach [38] and evaluated the performance over multiple runs. For each run, a piece of data was selected for testing, and the remainder was used for model training. The testing results across all data (i.e., 458 lung CT scans with 113 radiomics features and labeled semantic features) were averaged to obtain the global system performance. A comprehensive performance evaluation of the proposed CT2Rep approach along with other well-known methods is provided in Table 3. For machine learning-based methods, the authors of [39] stated that determining the best machine learning method for radiology application is a decisive step, and hence we compared our approach against naïve Bayes [38,40], k-nearest neighbors (KNN) [38], and eXtreme gradient boosting (XGB) [41]. In addition, we also included the following in our comparison: recurrent neural network (RNN) [42], an alternative multilayer perceptron model (MLP) [27,28], and convolutional neural network (CNN) [43,44], which are three deep neural network-based models.
(4)$$\mathrm{Precision}=\frac{TP}{TP+FP}$$
(5)$$\mathrm{Recall}=\frac{TP}{TP+FN}$$
(6)$$F_{1}=\frac{2\times \mathrm{Precision}\times \mathrm{Recall}}{\mathrm{Precision}+\mathrm{Recall}}$$

Table 3 displays the comparison results. As a baseline, the NB classifier with a basic statistical model only reached a mediocre performance. Since it only considers features using conditional probability, it is difficult to represent inter-feature relations. The overall F_1_-score of the NB classifier is 50.93%, and especially for the feature Margin, the F_1_-score is as low as 27%. Next, the KNN method outperforms NB simply by calculating data similarity in the feature space. This can be attributed to the distribution of data that could be distinctively divided and partially interdependent, resulting in the superior outcome of the KNN. We also looked at the XGB method, which is a gradient boosting decision tree that integrates multiple learners for classification problems. This mechanism effectively learns the discriminative feature set, thus greatly outperforming NB and KNN with an overall F_1_-score of 66.32%. As for neural network-based methods, since RNN is proven to be applicable to sequential processing and CNN is suitable for visual feature extraction [45,46], these neural models (i.e., RNN, MLP, and CNN) can further improve performances up to about 75%, 84%, and 85%, respectively. Nonetheless, the proposed CT2Rep method not only integrated multiple MLPs that can extract latent features from radiomics features, but also learned the relationship between location, margin, and texture of lung cancer through a joint learning scheme. Moreover, multiple dense layers are stacked to distil discriminative features for the purpose of identifying the target (i.e., semantic feature prediction). Consequently, the CT2Rep method achieves the best precision, recall, and F_1_-scores among all compared methods. In addition, we evaluated the performances of the compared methods by observing their ROC curves [38,47]. As shown in Figure 3A–C, the true positive rate of our method is superior to those of the compared methods in predicting all three semantic features. In other words, the CT2Rep method is effective in recognizing semantic features from lung CT scans. Finally, we evaluate the performances of the compared methods using the ROC curve on three semantic labels. To plot the curves, the evaluated instances were sorted according to their prediction scores with True Positive Rate and False Positive Rate. Figure 3A–C shows that the CT2Rep is superior to those of the compared methods. In other words, our method is able to accurately predict semantic labels behind CT scans.

To summarize, the proposed CT2Rep achieves a remarkable performance in recognizing semantic features by means of a multi-objective learning scheme. The proposed method successfully integrates the extracted radiomics features for jointly learning to predict Location, Margin, and Texture of lung cancer, which is why it can surpass the com-pared methods.

### 3.2. Satisfaction Analysis for Radiology Report Generation

Moreover, we constructed a prototype of the CT2Rep system, as illustrated in Figure 4. Given lung CT scans, the corresponding radiology reports can be automatically generated. We included the most important factors (i.e., Location, Texture, Size [48], and Margin) when composing a radiology report, as well as other useful information to recognize the degree of lung cancer Lung-RADS. The chart denoting the distribution of the scores in the satisfaction study is shown in Figure 5. The scores 4 and 5 accounted for 95% of the total evaluated data. This indicates that both radiologists were satisfied with the quality and accuracy of the report text, which validates the high quality of our automatic report and the practicality of this research. Nevertheless, there were still a few outliers with scores 2 and 3 in the satisfaction questionnaire, which means that radiologists were not that satisfied with these cases. On the basis of further analyses of the result, it was found that these cases were the result of incorrect predictions of the location feature. The radiologists were especially sensitive to the prediction of tumor location, and mistakes about this were deemed unacceptable. Furthermore, the incorrect Location prediction has an impact on the performance of Texture and Margin predictions. As a result, the Location information is very sensitive in the judgment of tumors by radiologists. Therefore, one of our directions for future research is to improve the accuracy of Location prediction by finding more useful information and features from the raw data.

## 4. Discussion

Radiology report generation through chest radiography interpretation is a time-consuming task that involves the examination of images by expert radiologists. It is common for fatigue-induced diagnostic error to occur, and especially difficult in areas of the world where radiologists are not available or lack diagnostic expertise. In this study, we demonstrated the effectiveness of different machine learning [49] and deep learning methods [50] for predicting the location, margin, and texture of lung cancer on the basis of radiomics features. Interestingly, the results were exceptionally different in terms of the margin prediction. The machine learning-based approaches generally performed worse, whereas the deep learning methods were able to achieve remarkable outcomes. In addition, it was also observed that texture prediction was comparatively less challenging than the others. All of the compared methods were able to obtain satisfactory results, including the NB and KNN baselines that were able to achieve about 80% F1-scores. The experiment illustrates that deep learning methods are more suitable for predicting semantic features using radiomics data of the lung CT scan. Integrating multiple deep neural networks to jointly model these characteristics can lead to further improvements. In addition, we validated the effectiveness of our model through a user satisfaction analysis. We found that, using our fully functional demo system, medical professionals can automatically obtain radiology reports from CT scans of the lungs. This can greatly alleviate the workload of radiologists and allow professional radiologists to devote more of their precious time and focus on examining images or conducting more long-term research [51].

Previous research on radiology report generation and Lung-RADS prediction is very scarce [18]. This work aims to bridge the gap and differs from existing research in the following aspects. First, we generated radiology reports from low-dose lung CT through transferring radiomics features to semantic labels, and further integrated the LungRADS features. To the best of our knowledge, this has not been conducted in previous research. Additionally, our proposed system also has its impact in public health. Deep learning approaches can be utilized to help simplify the diagnostic procedure and improve disease management, making a substantial contribution to public health purposes [52]. From the patient’s perspective, in addition to earlier detection and earlier treatment, such an auxiliary decision-making system can not only speed up the treatment process, but also advance the time point for discussing the prognostic status. It is crucial to note that the use of computers in image reading can be helpful in providing fast and accurate results but requires careful validation due to the impact on patients’ lives.

There are several notable limitations that should, nevertheless, be acknowledged. First, we only targeted patients with lung cancer. Thus, healthy people or those suffering from other kinds of cancer were not covered. Moreover, our samples were all selected from CT scans containing a single nodule. Therefore, our method currently can only identify and generate reports for CT scans in this category. Second, the effectiveness evaluation of the current method was made specifically for the extraction of radiomics features from 3D images. That is, if the input data are changed to 2D images, the recognition accuracy of the model remains understudied, and the effectiveness evaluation results of 2D images are currently not available. Last but not least, the purpose of designing this system is not to replace radiologists, because the generated report only contains the factors Location, Margin, Texture, Size, and Lung-RADS, which are not comprehensive enough for a thorough diagnosis. As stated in [53,54], a clinical decision system that can assist radiologists in making decisions is highly expected, and the author of [55] also concluded that computer-aided diagnosis would be used as an effective auxiliary tool in daily clinical diagnostic examinations. Moreover, some tumor-related information must still be added or labeled by radiologists, and the generated report must be confirmed by experts before becoming the final report. In short, our model can be used as an essential auxiliary tool for a more extensive system.

## 5. Conclusions

In this paper, we proposed a multi-objective learning neural network called CT2Rep, which is effective in predicting semantic features, location, margin, and texture on the basis of the radiomics features extracted from lung CT scans. The results demonstrate great potential in this model for the production of robust and reliable quantitative lung diagnosis reports. Given that medical personnel or even ordinary users can obtain important indicators simply by providing the lung CT scan to the system, we envision wide-spread application of the proposed framework.

## Figures and Tables

**Figure 1 jpm-12-00417-f001:**
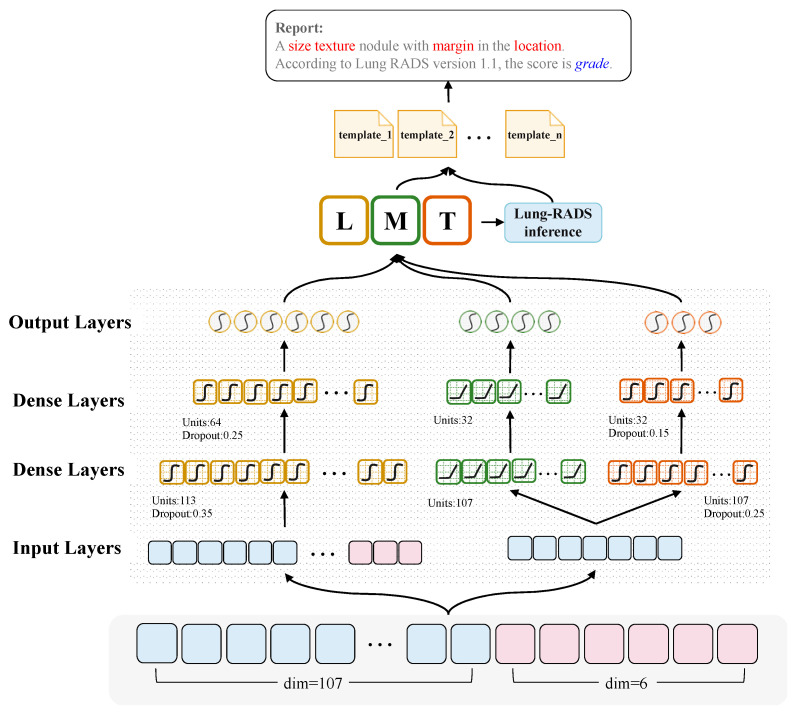
Illustration of the proposed multi-objective deep learning framework for semantic feature prediction. For prediction of the final targets: location (L), margin (M), and texture (T), the input layer of the proposed model is a multi-feature fusion vector that integrates 107-dimensional radiomics feature with 6 extra SISN features extracted from the information of nodule segmentation. Afterwards, there are two dense layers with dropout. The activation of the last two dense and the output layer are different, wherein ReLU is used for predicting Margin and tanh is used for Location and Texture. The final activation function of the output layer to predict results is softmax. Using the prediction results of these three semantic features, the radiology reports can be generated through filling slots of a set of predefined templates.

**Figure 2 jpm-12-00417-f002:**
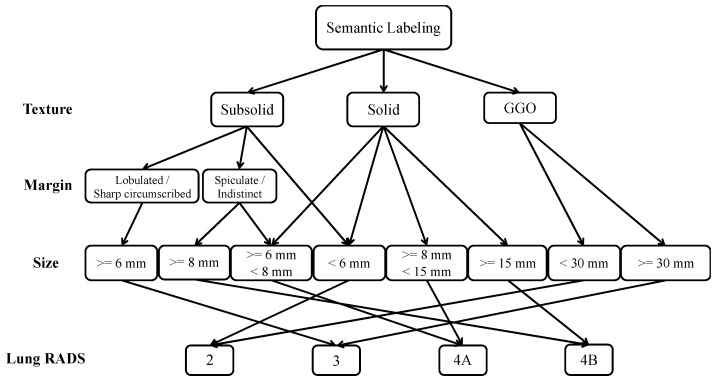
The decision process of the Lung-RADS model. The criterion for determining Lung-RADS at the first (top) level is the type of texture, including Subsolid, Solid, and GGO. Then, if the Texture is Subsolid, the category of margin is further categorized into Lobulated/Sharp Circumscribed, or Spiculate/Indistinct. The last criterion is the size, which denotes how big the tumor is, and the final definition of Lung-RADS categories depends on the size of the tumor.

**Figure 3 jpm-12-00417-f003:**
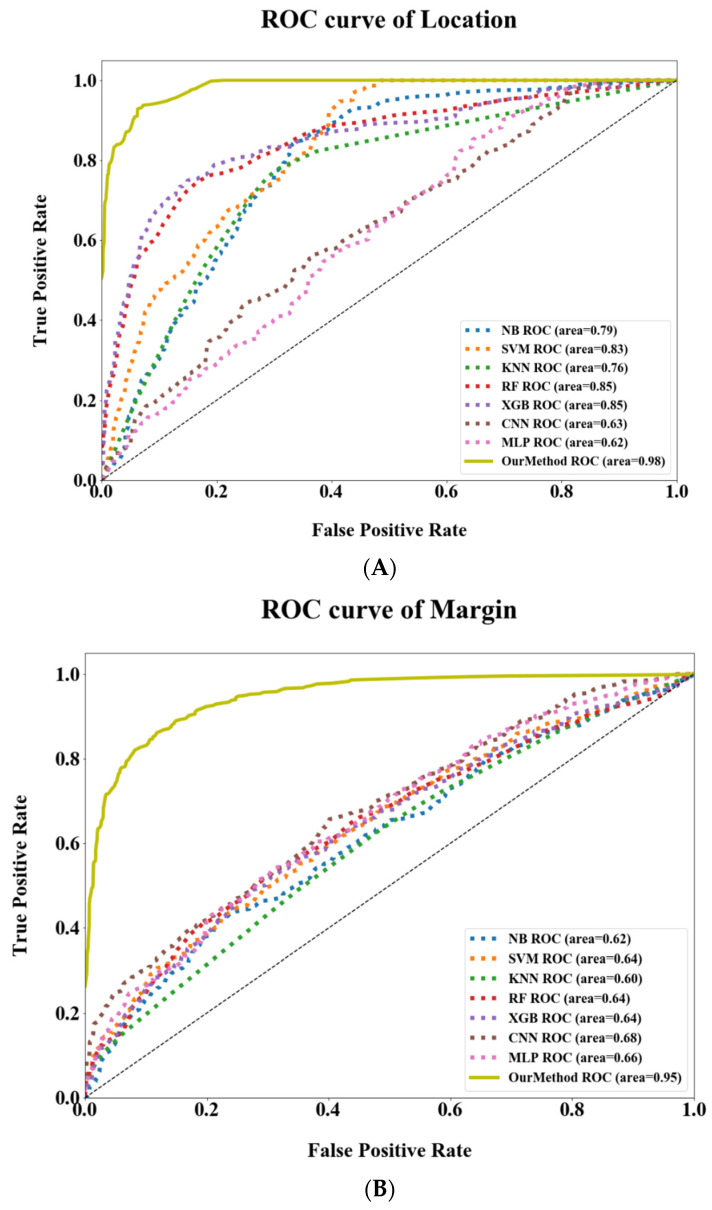
ROC Curves of the CT2Rep and compared methods for three semantic features: (**A**) Location, (**B**) Margin, and (**C**) Texture. The ROC curves with AUC include 8 approaches for predicting semantic features. The horizontal axis of these plots is the False Positive Rate, which also stands for probability of false alarm, whereas the vertical axis is True Positive Rate, which also stands for sensitivity.

**Figure 4 jpm-12-00417-f004:**
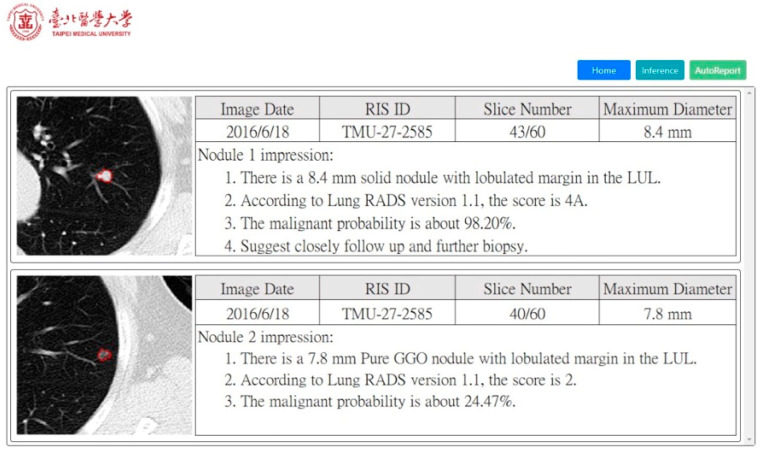
Illustration of the prototype image analysis and prediction system based on CT2Rep. On the left is the patient’s CT scan and segmented nodules. On the right are the basic metadata of the CT scan, as well as the generated descriptions, which integrates the information of Size, Location, Margin, Texture, and Lung-RADS. Here, two lung nodules were detected from the same patient. Noticeably, since these nodules were located in different slices and their characteristics were distinct from each other, different reports were generated accordingly. As demonstrated here, this system can effectively assist radiologists in discovering all suspected nodules first. Subsequently, false-positives will be judged and verified by radiologists.

**Figure 5 jpm-12-00417-f005:**
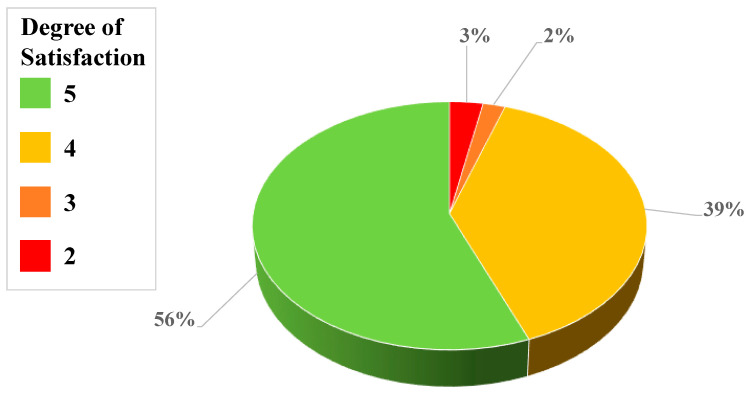
Distribution of satisfaction scores from the questionnaire by professional radiologists. The questionnaire is designed to collect feedback from the professionals regarding the experience when testing the automatic report generation system. The colors indicate different degrees of satisfaction, where green is used for higher scores (more satisfied) in the questionnaire, and red indicates the lower scores (less satisfied).

**Table 1 jpm-12-00417-t001:** List of labels for three types of semantic features.

Semantic Feature	Label
Location	RUL
	RML
	RLL
	LUL
	LLL
	Lingular Lobe
Texture	Solid
	Subsolid
	Pure GGO
Margin	Sharp Circumscribed
	Lobulated
	Indistinct
	Spiculated

**Table 2 jpm-12-00417-t002:** Amount of each feature type that was extracted.

Group		Feature	Amount	Total
Radiomics	First-order	Intensity	18	107
		shape-based	14	
	Second-order	GLCM	24	
		GLRLM	16	
		GLSZM	16	
		NGTDM	5	
		GLDM	14	
Segmentation		SISN	6	6

**Table 3 jpm-12-00417-t003:** The performance results of the compared methods.

Method	Location	Margin	Texture	Macro avg.
	Precision, Recall, F_1_-Score (%)			
NB	49.40/49.78/49.05	34.20/33.62/27.10	82.37/76.20/76.65	55.20/53.20/50.93
KNN	47.82/50.44/48.99	40.64/40.61/39.97	84.56/86.90/84.85	57.67/86.90/57.94
XGB	69.23/73.58/70.89	40.49/40.83/40.13	87.32/88.65/87.45	65.68/67.69/66.32
RNN	64.46/65.07/57.05	83.19/78.82/77.22	93.36/92.79/91.76	80.34/78.89/75.34
MLP	85.69/85.81/83.32	84.12/81.22/79.50	92.81/82.14/91.05	87.54/86.39/84.62
CNN	86.39/85.81/82.99	86.25/82.75/81.34	93.18/92.58/91.62	88.61/87.05/85.32
CT2Rep	85.68/87.12/84.66	83.36/83.41/83.10	94.11/94.10/94.09	87.72/82.74/87.29

## Data Availability

Data are available on request due to privacy.

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
