# Peer review of "Deep Multi-Objective Learning from Low-Dose CT for Automatic Lung-RADS Report Generation"

_jpm, 2022, doi:10.3390/jpm12030417_

Round 1

Reviewer 1 Report

Dear Author;

1.The satisfaction survey's scoring criteria should be mentioned in the material and method section. The results section should contain the data from the satisfaction survey, which should be moved from the discussion section.

2.On line 304, there is a paragraph error that needs to be fixed.

Author Response

Dear Reviewer,

Thank you for your thorough review of our paper, as well as constructive comments, corrections and suggestions. A major revision of the paper has been carried out, taking all of your comments into account, and we believe that in the process the paper has been significantly improved.

Response: Our major revision of the paper includes (i) an adjustment of sections to improve the readability and (ii) fixing a paragraph error. Our responses to each of your comments are listed below.

Point 1: The satisfaction survey's scoring criteria should be mentioned in the material and method section. The results section should contain the data from the satisfaction survey, which should be moved from the discussion section.

Response: Thank you for the suggestion. In this revised version, we have added a description to explain the scoring criteria of the satisfaction survey in the Material and Method section. Moreover, the content describing the data of the satisfaction survey has been moved to the Results section.

Point 2: On line 304, there is a paragraph error that needs to be fixed.

Response: Thank you for the comment. In this revised version, we have made the revision accordingly.

Reviewer 2 Report

The main topic covered by the article is the use of a multi-objective deep learning model called CT2Rep (Computed Tomography to Report) to generate chest radiology reports by extracting semantic features from chest CT scans. The results and conclusions are promising and demonstrate some applications that are important.

The subject is innovative for several reasons, both in terms of the extent of the research carried out and the areas covered and their application. It stands out from the other models cited in this research by the novelty of the relevant and very accomplished approach that has been put in place to obtain reliable results in the field of radiological report generation and interpretation of chest radiographs.

This study demonstrated the effectiveness of different machine learning and deep learning methods to predict the location, and to determine the margin and texture of lung cancer based on radiomic features. The results of the comparison with other methods are remarkable. This makes the approach chosen by based on deep machine learning.

The novelty and originality of this research relate to the wealth of references reviewed. These are established on the basis of a very relevant research strategy, in particular that which is cited in the bibliographies. The references are rich and adapted to the requirements of the subject.

The improvements envisaged in relation to established methodologies are obtained by comparing their performance. However, this deserves to be further explored by the authors, who may also consider further construction of the CT2Rep prototype, particularly around clearer and more concise reports that are presented to radiologists for interpretation, validation and acceptance.

The conclusions are relevant and consistent with the issues cited in the introduction.

Author Response

Point 1: The main topic covered by the article is the use of a multi-objective deep learning model called CT2Rep (Computed Tomography to Report) to generate chest radiology reports by extracting semantic features from chest CT scans. The results and conclusions are promising and demonstrate some applications that are important.

The subject is innovative for several reasons, both in terms of the extent of the research carried out and the areas covered and their application. It stands out from the other models cited in this research by the novelty of the relevant and very accomplished approach that has been put in place to obtain reliable results in the field of radiological report generation and interpretation of chest radiographs.

This study demonstrated the effectiveness of different machine learning and deep learning methods to predict the location, and to determine the margin and texture of lung cancer based on radiomic features. The results of the comparison with other methods are remarkable. This makes the approach chosen by based on deep machine learning.

The novelty and originality of this research relate to the wealth of references reviewed. These are established on the basis of a very relevant research strategy, in particular that which is cited in the bibliographies. The references are rich and adapted to the requirements of the subject.

The improvements envisaged in relation to established methodologies are obtained by comparing their performance. However, this deserves to be further explored by the authors, who may also consider further construction of the CT2Rep prototype, particularly around clearer and more concise reports that are presented to radiologists for interpretation, validation and acceptance.

The conclusions are relevant and consistent with the issues cited in the introduction.

Response: Thank you for your very careful review of our paper and constructive comments, corrections, and suggestions. A major revision of the paper has been carried out while taking all of your comments into account. We believe that, in this process, the paper has been significantly improved. As you have suggested, we are developing the CT2Rep prototype and in the process of deploying it at the Taipei Medical University Hospital.

Reviewer 3 Report

First of all, I would like to thank for the opportunity to review this paper. Extracting information from chest computed tomographic scans and converting it into medical reports can be time-consuming for radiologists and may improve the quality of healthcare. In this context, the paper under review is aimed at multi-objective deep learning model for generating lung radiology reports by extracting semantic features from lung CT scans.

The subject under study is certainly very important, especially in the historical period we are experiencing. The article presents interesting results but, but it is nevertheless believed that, given the organization of the contents and the description of the same, the manuscript cannot be published in its current form. I would like to encourage authors to consider several issues to be improved.

Title: it is too long and confusing. It should be shortened highlighting the object of the study.

Introduction: The authors should make clearer what is the gap in the literature that is filled with this study. The authors do not frame their study within the vast body of literature that addressed the issue of automatic diagnostic imagines reading. What is the possible contribution of the study to the literature? What are the implications of the study? The objectives should be better explained at the end of the section.

Methods: The survey was conducted using a pool of CT and developing a self-made model. The paper will benefit from a better description of the model. The enrolment procedure must be better specified. How did the authors choose the way to select the sample? This can represent a great bias origin. How did they avoid the selection bias? What is the reference population? what is the minimum sample size?

Ethical Issue: the number of the approval released by an Ethical competent body must be reported. The paper deal with patients personal data and diagnostic materials, a Review board is not enough.

Discussion: I also suggest expanding. What is the possible international contribution of the study to the literature? What are the implications of the study? Emphasize the contribution of the study to the literature. The discussion must be updated with one of the principal argument in this context: the possibility of an automatic reading of TC versus RX and its impact in public health (refer to articles with doi: 10.7416/ai.2021.2467).

Author Response

Dear Reviewer,

Thank you for your thorough review of our paper, as well as constructive comments, corrections and suggestions. A major revision of the paper has been carried out, taking all of your comments into account, and we believe that in the process the paper has been significantly improved.

Point 1: First of all, I would like to thank for the opportunity to review this paper. Extracting information from chest computed tomographic scans and converting it into medical reports can be time-consuming for radiologists and may improve the quality of healthcare. In this context, the paper under review is aimed at multi-objective deep learning model for generating lung radiology reports by extracting semantic features from lung CT scans. The subject under study is certainly very important, especially in the historical period we are experiencing. The article presents interesting results but, but it is nevertheless believed that, given the organization of the contents and the description of the same, the manuscript cannot be published in its current form. I would like to encourage authors to consider several issues to be improved.

Response: Thank you for your valuable and detailed comments on our work. They have helped us undertake a major revision of the paper that includes:

  • Improve readability and provide a more concise title.
  • Highlight the contributions of this work.
  • Data curation procedure.

We have replied to each of your comments in the following paragraphs.

Point 2: Title: it is too long and confusing. It should be shortened highlighting the object of the study.

Response: Thank you for the suggestion. In this revised version, we have made the revision accordingly.

Point 3: Introduction: The authors should make clearer what is the gap in the literature that is filled with this study. The authors do not frame their study within the vast body of literature that addressed the issue of automatic diagnostic imagines reading. What is the possible contribution of the study to the literature? What are the implications of the study? The objectives should be better explained at the end of the section.

Response: We thank you for pointing this out and making the suggestion to improve the readability of the Introduction section. In this revised version, we have added a description to discuss more literatures related to automatic diagnostic image reading in the past three years. After the literature review, we observe that only a few are focused on radiology report generation. Therefore, this highlights the value and contribution of our research, which targets at generating radiology reports from low-dose lung CT through transferring radiomics features to semantic labels. Notably, another novelty of this work lies in the integration of LungRADS predicted by our model into the generated report, which has not been done in previous research. We also included additional references that can help readers better comprehend this work, as well as highlight our contributions through more comparison with previous works.

Point 4: Methods: The survey was conducted using a pool of CT and developing a self-made model. The paper will benefit from a better description of the model. The enrolment procedure must be better specified. How did the authors choose the way to select the sample? This can represent a great bias origin. How did they avoid the selection bias? What is the reference population? what is the minimum sample size?

Response: Thank you for giving us the opportunity to clarify the procedure of selecting the sample. These patients were reviewed retrospectively from 2016 to 2019, collected from TMUH, Taipei Medical University Shuang Ho Hospital, and Taipei Municipal Wanfang Hospital. This indicates that the collected data spans over a long period of time and from multiple hospitals. Therefore, the bias of selected samples is minimal. We have explained this more clearly and made the appropriate revision.

Point 5: Ethical Issue: the number of the approval released by an Ethical competent body must be reported. The paper deal with patients personal data and diagnostic materials, a Review board is not enough.

Response: We appreciate this call for clarification. In the revised Materials and Methods section, we have added a more detailed description and explanation of the ethical issue. Regarding the legality of data acquisition, this study was approved by the Joint Institutional Review Board of the TMUH (IRB No. N202202003). The records were collected by the TMU Office of Data Science (TMUODS) from pathologically confirmed patients with final pathological confirmation or clinical diagnosis. TMUODS has been integrating electronic medical records of three affiliated hospitals of Taipei Medical University to form the Taipei Medical University Clinical Research Database (TMUCRD). With the establishment of the TMUODS and after obtaining the consent of the patients, Taipei Medical University de-identified the clinical data from these affiliated hospitals and constructed this clinical database. At the same time, it also formulated database management methods and application procedures to standardize the management of the database, access, usage rules, security protocol, and enable health data to be used for biomedical research. During the data collection period, approximately 400 people passed away, and 100 people opted-out.

Point 6: Discussion: I also suggest expanding. What is the possible international contribution of the study to the literature? What are the implications of the study? Emphasize the contribution of the study to the literature. The discussion must be updated with one of the principal argument in this context: the possibility of an automatic reading of TC versus RX and its impact in public health (refer to articles with doi: 10.7416/ai.2021.2467).

Response: Thank you for this suggestion. In this revised version, we refer to the article you mentioned and made the revision accordingly.

Round 2

Reviewer 3 Report

The Authors declare that "In this revised version, we refer to the article you mentioned and made the revision accordingly" but probably the authors did not submitt the final version since this point was not addressed and the reference is missing.

Author Response

Point 1: The Authors declare that "In this revised version, we refer to the article you mentioned and made the revision accordingly" but probably the authors did not submitt the final version since this point was not addressed and the reference is missing.

Response: Thank you for your thorough review of our paper, as well as constructive comments, corrections, and suggestions. A minor revision of the paper has been carried out, taking all of your comments into account, and we believe that in the process the paper has been significantly improved. In this revision, we have added the reference which you suggested as index 52.
